# OpenReview forum: "SafeEvalAgent: Toward Agentic and Self-Evolving Safety Evaluation of LLMs"
_ICLR.cc/2026/Conference — ICLR 2026 Conference Withdrawn Submission_

### Official Review · Reviewer_xWnL · 2025-10-25

**Soundness:** 3
**Presentation:** 3
**Contribution:** 3
**Rating:** 6
**Confidence:** 4

**Summary:**

This paper introduces SafeEvalAgent, a multiagent framework for continuous, self-evolving LLM safety evaluation. Addressing limitations of static benchmarks, it autonomously processes unstructured policy docs to build a testable knowledge base and generate evolving test suites via a synergistic pipeline (Specialist, Generator, Evaluator, Analyst agents). Experiments on 11 LLMs (e.g., GPT-5, Gemini-2.5) across 3 regulations (EU AI Act, NIST RMF, MAS FEAT) show declining safety rates with harder tests (e.g., GPT-5’s EU AI Act rate drops from 72.50% to 36.36%), proving its ability to uncover deep vulnerabilities missed by traditional methods.

**Strengths:**

First, it addresses static benchmark flaws by enabling continuous, self-evolving evaluation to adapt to dynamic AI risks and regulations .

Second, its multiagent pipeline autonomously processes unstructured policies into testable knowledge, aiding targeted test generation .

Third, experiments confirm it uncovers deep LLM vulnerabilities missed by traditional methods, with reliable evaluator-agent judgments

**Weaknesses:**

First, it lacks exploration of how the framework performs with more diverse global regulatory documents beyond the three tested.

Second, it should supplement more evaluation data to verify the proposed method’s generalization across broader LLM types and scenarios.

Third, it doesn’t discuss potential computational cost trade-offs when scaling the self-evolving loop to more iterations.

**Questions:**

see Weaknesses

---

### Official Review · Reviewer_91cv · 2025-10-27

**Soundness:** 3
**Presentation:** 3
**Contribution:** 1
**Rating:** 2
**Confidence:** 5

**Summary:**

The paper proposes SafeEvalAgent, a multi-agent, self-evolving framework for continuously evaluating LLM safety. It transforms regulatory documents (e.g., EU AI Act, NIST AI RMF) into structured rules, generates diverse test suites through autonomous agents, and iteratively refines evaluations via a feedback loop that learns model weaknesses. Experiments on 11 LLMs (GPT-5, Gemini, Llama, Qwen, etc.) show large safety drops across iterations, revealing hidden vulnerabilities missed by static benchmarks. The work demonstrates a shift from static audits to dynamic, regulation-grounded safety evaluation.

**Strengths:**

1. Comprehensive evaluation on the SOTA LLMs.
2. Demonstrate the effectiveness of the proposed pipeline, and the self-evolving evaluation loop can indeed help discover more vulnerabilities of the target LLM

**Weaknesses:**

Main concern:
- Lacks sufficient novelty and miss some related literature. The concept of agent-based evaluation has been explored in prior studies, and the idea of a self-evolving evaluation loop is not new. However, the authors do not cite or discuss the relevant works [1], which weakens the paper’s positioning and contribution. Furthermore, the additional components, the regulation-to-knowledge transformation and the test suite generation, are relatively incremental and offer limited methodological innovation. Thus, this paper lacks novelty.

Other concerns
1. How do author ensure the $Q^{k+1}$ share the same semantics as $Q^{0}$? Is there a metric to quantitatively evaluate how much $Q^{k+1}$ still captures the essence of $Q^{0}$? Also, is there a metric to measure the semantic relationship between the $q_m$ and $q_{base}$? Evaluating the semantic fidelity of rephrased questions is essential to ensure that the generated variants remain faithful to the intended rule rather than drifting from the original meaning.
2. The paper should include metrics to assess how effectively the proposed framework converts regulatory text into structured knowledge. Moreover, it would be valuable to evaluate the rationales behind the rules in the leaf nodes: do these indeed correspond to misbehaviors or policy mentioned in the source documents?
3. When the model is asked to generate the adversarial perturbation, would the model refuse to follow the instructions to generate such perturbations? Also, how to ensure the q and c are correct in the deterministic probes and multimodel grounding?

[1] Ali-agent: Assessing llms' alignment with human values via agent-based evaluation. NeurIPS 2024.

**Questions:**

1. What is the cost of this evaluation pipeline?

---

### Official Review · Reviewer_DMSx · 2025-10-28

**Soundness:** 2
**Presentation:** 3
**Contribution:** 3
**Rating:** 4
**Confidence:** 4

**Summary:**

This paper proposes SafeEvalAgent, a multi-agent system, that automatically interprets regulatory documents and generates LLM safety assessment benchmarks. The system consists of four specialized agents. The Specialist Agent structures regulations into a hierarchical knowledge base and enriches it with real-world examples, while the Generator Agent generates various types of question groups (basic, jailbreak, multimodal, MCQ, etc.). The Evaluator Agent evaluates model responses using rubric-based judgments, and the Analyst Agent analyzes failure patterns and generates refined attack strategies. Moreover, "self-evolving" evaluation of SafeEvalAgent uncovers deeper vulnerabilities through iterative refinement, and for building a "continuous" and "regulation-grounded" evaluation ecosystem. The experiments evaluated 11 state-of-the-art models (five proprietary models, including GPT-5, Gemini-2.5, and Grok-4, and six open-weight models, including Qwen, Llama, and DeepSeek) against three regulations (EU AI Act, NIST AI RMF, and MAS FEAT). A consistent decrease in security was observed across all models with each iteration.

**Strengths:**

1. This paper establishes a practical end-to-end automated pipeline for regulation-grounded safety evaluation. It receives regulatory PDF documents as input, generates a structured knowledge base, and builds a test suite based on this, fully automating the entire process, from initial processing to the final evaluation report. Compared to existing research like COMPL-AI and AutoLaw, which required manual codification, this system offers practical value by enabling rapid response to new regulations.

2. The large-scale empirical study includes 11 diverse models against three distinct regulatory frameworks (the legally binding EU AI Act, the voluntary NIST AI RMF, and the financial domain-specific MAS FEAT). The paper provides a comprehensive analysis. Notably, all models consistently exhibited reproducible vulnerability discovery patterns, and the finding that even GPT-5 experienced a 36.4% decrease in security provides a strong warning to the industry.

3. It sets an example in transparency and reproducibility. The authors provide detailed agent prompts in the Appendix, illustrate the entire process through an end-to-end case study, and achieve 88-91% agreement and Cohen's Kappa of 0.77-0.81 through human validation.

**Weaknesses:**

1. Major concern on the paper is that its claim of "self-evolving" doesn't align with the actual implementation. While the paper uses terms like "continuous," "self-evolving," and "living ecosystem," the system actually terminates after a fixed number of three iterations. If a new regulation comes, it requires a restart from scratch, and continual updates aren't supported. This framework is self-refinement that obtains question with higher safety risk, not a self-evolving framework.

2. The title of the paper, "Safety Evaluation of LLMs," is very general, but it actually focuses on "Regulation Compliance Checking." General toxicity and bias issues are addressed only within the scope of regulations, and general safety aspects such as robustness and privacy are not included. It is crucial to clearly define the scope and actual contributions of the paper.

3. Key component-wise ablation studies are missing. The study did not separately measure the effects of using structure alone, adding enrichment, adding search-augmentation, or applying iteration. The semantic similarity heatmap in Figure 2 demonstrates that the structured knowledge base is semantically consistent, but it fails to demonstrate whether this translates into actual performance benefits.

4. Multimodal testing is a key feature of the paper, but quality verification related to image generation was not conducted at all. Only a single example is presented in Table 8, and human validation only reported overall judgment accuracy (88-91%). Various quality issues can arise, including failed or inappropriate image generation, insufficient inclusion of regulatory violations, and misalignment of images and text.

5. Search-augmented enrichment appears effective for the present (2024-2025), but concerns exist about long-term scalability. Currently, searching for "EU AI Act examples" yields fresh and relevant results, but problems could arise after 2030 as regulations become more complex and more complex.

6. Test diversity analysis was not systematically conducted. Repeated refinement and pattern analysis for attack risk reducing question diversity. Diversity within question groups (intra-group diversity), the extent to which tests for different rules overlap (inter-group redundancy), and which regulatory areas were not sufficiently tested (coverage gaps) were not measured. Only safety rates were measured, and there was no quantitative analysis of the diversity and coverage of the generated tests.

**Questions:**

Please refer to the weakness above.

**Details Of Ethics Concerns:**

Although authors properly include the ethical statement on this paper. Some potentially harmful applications need to be discussed. The system includes the tools regarding jailbreaking, which can be utilized to attack LLMs.

---

### Official Review · Reviewer_r4BY · 2025-10-31

**Soundness:** 2
**Presentation:** 2
**Contribution:** 2
**Rating:** 2
**Confidence:** 4

**Summary:**

This paper proposes SafeEvalAgent, a multi-agent framework for dynamic and self-evolving safety evaluation of large language models. It autonomously converts regulatory documents into structured knowledge, generates multimodal safety tests, and iteratively refines them through a self-improving evaluation loop. Experiments across major LLMs reveal that model safety performance declines significantly as the evaluation hardens, exposing vulnerabilities missed by static benchmarks.

**Strengths:**

1. The paper contributes a agentic pipeline that autonomously transforms real-world regulations and policies into safety evaluation tests, which is conceptually valuable and practically relevant for the safety community.

2. The experimental evaluation is broad and thorough, covering multiple major LLM families across different regulatory frameworks, providing convincing empirical support.

**Weaknesses:**

1. Limited novelty.

The central idea that static safety benchmarks are insufficient and must be replaced by dynamic, agentic evaluation frameworks has already been extensively articulated in prior work such as ALI-Agent [1]. The present paper largely reuses this conceptual foundation, differing mainly in datasets and framing, which limits its contribution to methodological novelty.

2. Insufficient human validation and reliability assurance.

The paper includes only a small-scale human-in-the-loop check for the Evaluator agent, which confirms alignment with human judgment. However, no human verification is conducted for other crucial components such as the Specialist, Generator, or Analyst agents. As a result, the reliability and factual correctness of the automatically generated rules, test cases, and analyses remain uncertain. This weakens confidence in the overall trustworthiness of the proposed evaluation pipeline.

3. Lack of qualitative analysis.
The paper reports large numerical drops in safety scores to argue that its dynamic evaluation reveals hidden vulnerabilities, yet it provides no concrete examples or case studies illustrating these failures. Without textual demonstrations of specific unsafe outputs that static benchmarks miss, the results remain abstract and difficult to interpret, limiting the reader’s understanding of what kinds of risks the framework actually uncovers.

4. Self-Referential Evaluation.
All major components of the framework are powered by large language model agents, often from the same or similar model families as those being evaluated. Such circular evaluation introduces systematic bias and undermines the validity of the reported safety degradation, as it remains unclear whether the framework is uncovering genuine vulnerabilities or simply artifacts of model-model interactions.

References.
[1] Zheng et al. "Ali-agent: Assessing llms' alignment with human values via agent-based evaluation." Advances in Neural Information Processing Systems 37 (2024).

**Questions:**

1. The paper does not evaluate the comprehensiveness of its regulation-to-question transformation. How can the authors ensure that the generated safety tests truly cover the full scope and nuance of each regulation, rather than only partial or surface-level interpretations?

2. Could the authors clarify what substantive novelty their framework introduces beyond ALI-Agent?

---

### Note · Authors · 2025-11-12

I have read and agree with the venue's withdrawal policy on behalf of myself and my co-authors.